# Impact of a sanitation intervention on quality of life and mental well-being in low-income urban neighbourhoods of Maputo, Mozambique: an observational study

Ian Ross [1,2] Giulia Greco,[1] Zaida Adriano,[3] Rassul Nala,[4] Joe Brown,[5] Charles Opondo,[6,7] Oliver Cumming[2]

For numbered affiliations see end of article.

**Correspondence to**
Dr Ian Ross;
ian.ross@lshtm.ac.uk

## ABSTRACT

**Objectives** Toilet users often report valuing outcomes such as privacy and safety more highly than reduced disease, but effects of urban sanitation interventions on such outcomes have never been assessed quantitatively. In this study, we evaluate the impact of a shared sanitation intervention on quality of life (QoL) and mental well-being.

**Design** We surveyed individuals living in intervention and control clusters of a recent non-randomised controlled trial, and used generalised linear mixed regression models to make an observational comparison of outcomes between arms.

**Setting** Low-income unsewered areas of Maputo City, Mozambique.

**Participants** We interviewed 424 participants, 222 from the prior trial's intervention group and 202 from the control group.

**Interventions** The control group used low-quality pit latrines. The intervention group received high-quality shared toilets, with users contributing 10%–15% of capital cost.

**Outcomes** Our primary outcome was the Sanitation-related QoL (SanQoL) index, which applies respondent-derived weights to combine perceptions of sanitation-related disgust, privacy, safety, health and shame. Secondary outcomes were the WHO-5 mental well-being index and a sanitation Visual Analogue Scale.

**Results** The intervention group experienced a 1.6 SD gain in SanQoL compared with the control group. This adjusted SanQoL gain was 0.34 (95% CI 0.29 to 0.38) on a 0–1 scale with control mean 0.49. Effect sizes were largest for safety and privacy attributes. Intervention respondents also experienced a 0.2 SD gain in mental well-being. The adjusted gain was 6.2 (95% CI 0.3 to 12.2) on a 0–100 scale with control mean 54.4.

**Conclusions** QoL outcomes are highly valued by toilet users and can be improved by sanitation interventions. Such outcomes should be measured in future sanitation trials, to help identify interventions which most improve people's lives. Since SanQoL weights are derived from respondent valuation, our primary result can be used in economic evaluation.

## STRENGTHS AND LIMITATIONS OF THIS STUDY

⇒ We achieved balance on observable characteristics by enrolling individuals living in intervention and control clusters of a recent non-randomised controlled trial.
⇒ Because intervention compound residents previously shared a low-quality toilet in the same location, the mechanism driving our results is likely to be the characteristics of intervention toilets.
⇒ No previous study has evaluated the impact of an urban sanitation intervention on outcomes such as user-reported privacy, safety or mental well-being.
⇒ Limitations include the absence of preintervention outcome data, risk of bias from the eligibility criterion of still using the type of toilet consistent with intervention/control status.

## INTRODUCTION

Nearly two billion people globally lack access to 'basic' sanitation.[1] This deficit leads to 400 000 deaths from diarrhoeal disease annually, as well to helminth infections and other diseases.[2] However, inadequate sanitation has negative consequences beyond infectious disease, including for perceived outcomes such as privacy, safety and dignity.[3–5] These outcomes are considered aspects of quality of life (QoL) under the capability approach,[6] since they capture what people can be and do. Furthermore, these outcomes map onto regularly defined features of QoL in general, such as health, personal security and environmental conditions.[7 8] Sanitation-related QoL (SanQoL) is then defined as the subset of overall QoL which is directly affected by sanitation practices or services.[9]

Many factors are hypothesised to moderate effects of sanitation interventions on QoL outcomes.[5 9] Some of these factors relate to individual characteristics. For example,

women and girls might be at greater risk of infringements to their sanitation-related safety and privacy than men and boys.[4 10 11] People with reduced mobility such as older or disabled people may be more likely to fear falling into a pit latrine.[12 13] Other factors relate to the environment. For example, someone using a nearby toilet in an urban neighbourhood perceived as unsafe may feel less secure using it at night than someone in a rural area.[14]

In studies of what toilet users most value about sanitation, QoL outcomes such as privacy or status are usually high up their list, and often above disease prevention.[15–17] Therefore, expected QoL payoffs from household sanitation investments are important determinants of whether the public good of an excreta-free environment is achieved.[18 19] Since different sanitation interventions plausibly improve QoL to different extents, measurement of such outcomes in sanitation impact evaluations may help identify the most effective interventions.

However, infectious disease outcomes have often been the focus of sanitation impact evaluations.[20] A systematic review of qualitative studies of sanitation's relationship with mental and social well-being identified innovation in quantitative measurement of such outcomes as a research priority.[4] One quantitative study has explored the association in the general population between urban sanitation access and mental well-being outcomes,[21] and another has done so in rural areas.[22] No studies have quantified the broader QoL effect of a specific urban sanitation intervention, with one such study ongoing,[23] and another undertaken in a rural area.[24] In this study, we evaluate the effect of a shared urban sanitation intervention on SanQoL and mental well-being in urban Mozambique, as compared with existing use of shared pit latrines.

## METHODS
### Setting
In Mozambique, only 37% of the population has access to basic sanitation as defined by WHO/UNICEF.[1] Maputo city, Mozambique's capital, has a population of 1.1 million, with the majority living in basic settlements with unpaved roads.[25] Pit latrines are used by 41% of people, and less than half of faecal waste is safely managed.[26] Our study site comprises low-income neighbourhoods in a 10 km² area of the Nhlamankulu district (further detail and maps in online supplemental appendix A). In this area, the poorest people live in informally walled 'compounds' with many households sharing the same toilet. Though 99% have access to on-premises piped water, low-quality pit latrines are common. Such latrines often have unlined pits and squatting slabs made of wood or tyres, and no water seal (u-bend) providing a barrier to smells and flies. Privacy is a challenge since latrine walls are usually made with scrap corrugated iron or plastic sheeting, often with gaps and holes. Doors are makeshift and roofs uncommon.

### Study design
We report an observational study sampling households from the intervention and control clusters of a prior non-randomised trial with a controlled before-and-after design (ClinicalTrials.gov, NCT02362932).[27] Intervention compounds in the Maputo Sanitation trial (MapSan) were identified in 2015–2016 using the following criteria: (1) inhabitants sharing poor-quality sanitation; (2) at least 12 inhabitants; (3) inhabitants willing to contribute financially to construction; (4) legal on-plot piped water; (5) located in predefined neighbourhoods; (6) sufficient space for construction; (7) accessible for transportation of construction materials and (8) water table low enough for septic tank installation.

The MapSan trial enrolled the compound if there was at least one resident child younger than 48 months. As each intervention compound was enrolled, investigators concurrently enrolled a control compound according to criteria 1–4 above and by number of inhabitants (cluster size). Control compounds were located in the same or adjacent neighbourhoods. MapSan concluded that the intervention had no effect on any measure of child health, with a 24-month diarrhoea prevalence ratio of 0.84 (95% CI 0.47 to 1.51).[7]

### Participants
Eligible participants for our study were people aged 18 or over and: (1) living in MapSan intervention or control compounds for 4 or more years, since before the intervention; (2) still using the type of toilet consistent with intervention/control status (eg, pit latrine if control). The first criterion ensured that, prior to the intervention, all our participants had been using a pit latrine without a water seal in that same compound they still lived on. This aimed to reduce risk of selection bias, because there had been migration out of and into MapSan-enrolled compounds since 2015. The second criterion was motivated by the knowledge that an unknown number of control compounds had: (A) received non-government organisation (NGO) sanitation interventions under a post-MapSan programme or (B) autonomously upgraded their toilets. This criterion aimed to ensure a sufficient sample of people using low-quality toilets for the purposes of validity assessments reported elsewhere,[28] in the context of unknown levels of subsequent intervention and upgrading.

We aimed to recruit two people per compound (one man, one woman) from different households. We used trial records to preassess eligibility for the 593 MapSan compounds (clusters), leading to the exclusion of 35 (figure 1). The two lists of remaining MapSan intervention and control compounds were then randomly reordered, and visited in that new order. Procedures for sampling individuals within a compound are in online supplemental appendix B. They are summarised as inspection of the toilet used to assess eligibility, followed by listing of eligible individuals and then random sampling. A team of four fieldworkers interviewed participants in Portuguese,

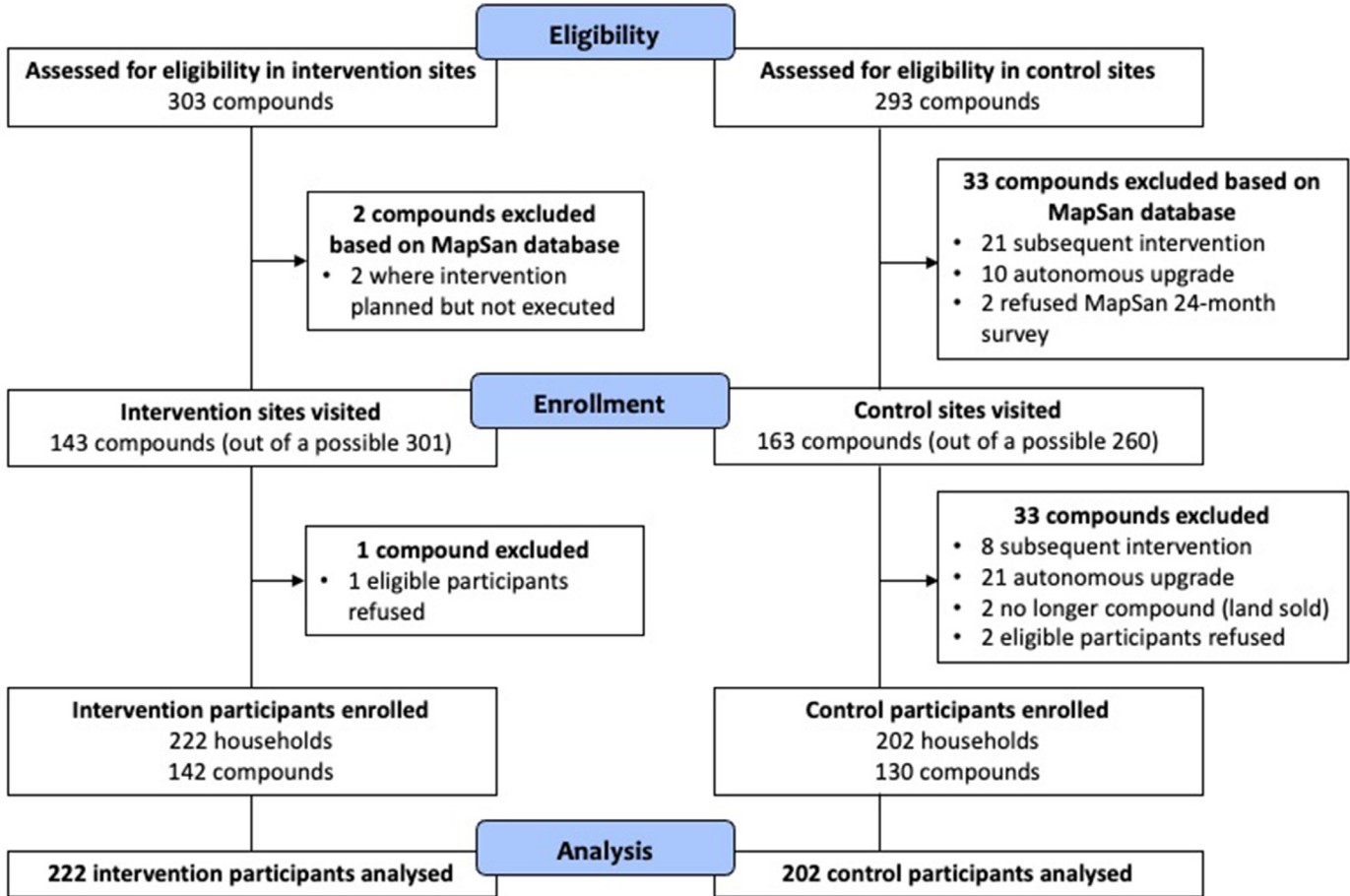

**Figure 1** Participant flow diagram showing eligibility, enrolment and analysis.

unless the participant preferred to talk in Changana, a local language in which all interviewers were fluent.

### Interventions

The intervention we evaluated was implemented by Water & Sanitation for the Urban Poor, an NGO. Compounds were provided with a subsidised pour-flush toilet with a water seal and concrete superstructure, discharging to a septic tank with soakaway (photos and further intervention details in online supplemental appendix A). Two toilet types were provided, depending on user numbers. A shared toilet (ST) with one stance (cubicle) was designed for around 15 people, while a community sanitation block (CSB) with two stances was designed for at least 21 people. Both STs and CSBs had metal doors lockable from the inside. Compound inhabitants had to pay a 10%–15% capital contribution—approximately US$120 for CSB (2015 prices) and US$80 for ST.

### Outcomes

The primary outcome is an index of SanQoL, deriving from a capability-based questionnaire informed by qualitative research.[9 28] SanQoL measures aspects of self-perceived QoL which are directly affected by sanitation practices or services. Validity and reliability of SanQoL were previously assessed in the Maputo setting through cognitive interviews and psychometric analysis.[28] The five

SanQoL attributes are disgust, health, privacy, safety and shame, measured on a four-level frequency scale (table 1). Responses are combined as an index by weighting attributes according to their relative importance, assessed via a ranking exercise undertaken with all study participants (online supplemental appendix B). The ensuing weights, which sum to 1 (table 1), were used to calculate SanQoL index values on a 0–1 scale. Higher scores are better, with 0 representing 'no sanitation-related capability' and 1 'full sanitation-related capability'. Histograms of outcome variables by group are in online supplemental appendix C.

The second outcome is a sanitation Visual Analogue Scale (VAS). We asked people to indicate on a paper-based 0–10 scale how they felt about their 'level of sanitation today', where 0 is 'worst imaginable sanitation' and 10 is 'best imaginable sanitation' (online supplemental appendix B). The rationale for including the sanitation VAS was to explore whether an effect size comparable to that for SanQoL would be seen when people rated their level of sanitation directly rather than via questionnaire items.

The third outcome is the WHO-5 mental well-being index, a multiattribute instrument for assessing subjective mental well-being.[29] It comprises five items related to feeling cheerful, calm, active, well rested and finding

**Table 1** SanQoL attributes and weights

| Attribute | Psychometric item | Responses | Weight in index valuation |
|---|---|---|---|
| Disgust | Can you use the toilet without feeling disgusted? | Always Sometimes Rarely Never | 0.22 |
| Health | Can you use the toilet without worrying that it spreads diseases? | | 0.29 |
| Privacy | Can you use the toilet in private, without being seen? | | 0.20 |
| Shame | Can you use the toilet without feeling ashamed for any reason? | | 0.13 |
| Safety | Are you able to feel safe while using the toilet? | | 0.16 |

In estimating index values, attribute-level scores are applied as 'always'=3, 'never'=0, etc. (formulae in online supplemental appendix B). SanQoL, Sanitation-related Quality of Life.

enjoyment in daily life, scored on a frequency scale (online supplemental appendix B). Scores are summed with equal weighting and rescaled to 0–100, with higher scores better. The rationale for including WHO-5 was that mental well-being is thought to be influenced by sanitation but, unlike SanQoL, is not specific to sanitation.[4]

### Hypotheses

We analysed participants according to trial arm, to test the overarching hypothesis that the intervention was associated with an improvement in QoL. Specific hypotheses were, first, that the intervention would be associated with higher SanQoL index values and sanitation VAS scores, because better-quality toilets have the potential to improve people's sanitation-related capabilities.[9 14] Second, we hypothesised that the intervention would be associated with higher mental well-being (WHO-5) scores, based on qualitative evidence[4] and earlier cross-sectional studies.[21 22]

We carried out exploratory moderation analyses for which the study was not powered. We assessed the hypotheses that for all three outcomes any effect would be larger for women than men,[30 31] and larger for elderly people (aged 60+)[32] than non-elderly.[12]

### Statistical analyses

The sample size calculation for the number of participants to be surveyed was estimated according to a formula for the comparison of two means with 80% power and significance at 0.05. The required sample size to detect a 0.05 mean difference in SanQoL with an SD of 0.15 and intracluster correlation coefficient of 0.4 was estimated as 398. We computed a wealth index using principal components analysis,[33] using the asset list from the most recent Mozambican Demographic and Health Survey. P values less than 0.05 were considered statistically significant evidence of association. We ran analyses in Stata V.17.[34]

To test hypotheses, we used generalised linear mixed models (GLMM), with gaussian distribution and identity link. Analyses were not prespecified. The model for SanQoL index values was as follows, with other hypotheses tested using the same model structure but a different dependent variable.

$$S_{ij} = \alpha_0 + \alpha_1 T + \beta X_{ij} + u_i + \varepsilon_{ij}$$

where:

$S_{ij}$ represents the SanQoL index value for individual $j$ in compound $i$.

$T$ is 1 for intervention and 0 for control.

$X_{ij}$ is a vector of covariates.

$\alpha_0$ is a constant with no interpretation in this case.

$\alpha_1$ is a coefficient and $\beta$ a vector of coefficients.

$u_i$ is a random effect at the compound level.

$\varepsilon_{ij}$ is the error term.

SEs were clustered at the compound level, since the intervention was delivered at this level, requiring the assumption that errors are not correlated across compounds. Spatial distribution of compounds was within one small area of Maputo (map in online supplemental appendix A). We included two types of covariates in $X_{ij}$. First, we adjusted for characteristics which were unbalanced at the 5% level between groups (table 2), that is, the wealth index only. Second, we included binary variables for gender and being elderly (aged 60+), because they are considered predictive of the participant's response to the intervention (as hypothesised in moderation analyses outlined above).[35 36] Only two participants had missing data for outcomes or covariates (one for WHO-5, one for the wealth index).

To test the hypothesis that intervention effects would be larger for women than men, we included a factorial interaction with $T$ for the gender variable. To test the hypothesis that intervention effects would be larger for older people, we included a factorial interaction with $T$ for the aged 60+ binary variable. As an additional exploratory analysis, we assessed effects on each of the five SanQoL attributes individually, by regressing on their raw scores (ranging 0–3). The rationale was to explore whether larger effect sizes were seen on some dimensions than others.

We assessed the sensitivity of results as follows. First, we included in $X_{ij}$ only covariates significantly different between groups at the 10% level (table 2) and excluded the gender and aged 60+ binary variables. Second, we included additional covariates hypothesised as predicting SanQoL and VAS (as well as gender and being aged 60+): whether the dwelling was rented, the number of people sharing a toilet stance, and whether the toilet was shared with other households. Third, we included

**Table 2** Characteristics of sample

| | Control (n=202) | Intervention (n=222) | P value for difference (t-test) |
|---|---|---|---|
| **Demographic characteristics** | | | |
| Respondent is male | 101 (50%) | 103 (46%) | 0.459 |
| Respondent age | 38.4 (14.9) | 41.2 (15.6) | 0.059* |
| Respondent aged 60+ | 23 (11%) | 32 (14%) | 0.355 |
| Respondent has a partner | 107 (53%) | 107 (48%) | 0.327 |
| Household size | 5.0 (2.8) | 5.2 (3.2) | 0.405 |
| No of children under 14 | 1.4 (1.5) | 1.2 (1.6) | 0.122 |
| **Wealth index** | | | |
| Wealth index score | −0.13 (1.00) | 0.12 (0.99) | 0.010** |
| *Dwelling has cement or tiled floor* | 184 (91%) | 210 (95%) | 0.160 |
| *Dwelling has concrete exterior walls* | 140 (69%) | 143 (64%) | 0.287 |
| *Access to electricity connection* | 167 (83%) | 192 (86%) | 0.277 |
| *Access to piped water connection* | 199 (99%) | 217 (98%) | 0.563 |
| *Household cooks indoors* | 114 (56%) | 114 (51%) | 0.295 |
| *Household owns television* | 153 (76%) | 184 (83%) | 0.069* |
| *Household owns fridge* | 98 (49%) | 128 (58%) | 0.060* |
| *Household owns mobile phone* | 166 (82%) | 191 (86%) | 0.278 |
| *Household owns bicycle* | 7 (3%) | 6 (3%) | 0.656 |
| *Household owns radio* | 63 (31%) | 96 (43%) | 0.010** |
| *Household owns watch* | 89 (44%) | 130 (59%) | 0.002*** |
| **Other respondent characteristics** | | | |
| Respondent completed primary school or above | 128 (63%) | 140 (63%) | 0.949 |
| Respondent completed secondary school or above | 18 (9%) | 33 (15%) | 0.060* |
| Respondent has moderate problems walking about, or worse | 12 (6%) | 13 (6%) | 0.971 |
| Respondent has moderate pain or discomfort, or worse | 21 (10%) | 17 (8%) | 0.325 |
| Respondent rents dwelling | 60 (30%) | 54 (24%) | 0.213 |
| Respondent's dwelling has zinc or concrete roof | 202 (100%) | 222 (100%) | n/a |
| Compound-level water & sanitation characteristics | | | |
| Water available at least 8 hours/day | 99 (49%) | 110 (50%) | 0.912 |
| Uses on-plot toilet | 197 (98%) | 219 (99%) | 0.397 |
| Shares toilet with other household(s) | 181 (90%) | 196 (88%) | 0.667 |
| No of households sharing stance | 3.3 (1.7) | 3.2 (1.6) | 0.511 |
| No of people sharing stance | 11.8 (5.2) | 12.6 (6.6) | 0.170 |

Data are n (%) for categorical variables and mean (SD) for numerical variables. *, **, *** indicate significance at the 10%, 5% and 1% level. Variables included in the wealth index are italicised. One participant had missing data for the wealth index. In the replication dataset, we categorised age, household size and children under 14 to maintain full anonymity, since several values were shared by five people or fewer. This table reports the mean of continuous values.

additional covariates hypothesised as predicting mental well-being: having a partner, being in moderate pain, or having moderate problems walking. Fourth, we explored whether using a GEE or ordinary least squares (OLS) specification instead of GLMM affected the results. We include the Strengthening the Reporting of Observational Studies in Epidemiology (STROBE) checklist,[37] as well as a reflexivity statement[38] (online supplemental appendix D).

### Ethical approval

The study received prior approval from the *Comité Nacional de Bioética para a Saúde* (IRB00002657) at the Ministry of Health in Mozambique, and the Research Ethics Committee of the London School of Hygiene and Tropical Medicine (Ref: 14609). Informed, written consent was obtained from all participants. Data have been published in accordance with consent.[39]

## Patient and public involvement

Members of the public were not involved in the design or conduct of this specific study. However, members of the public were involved in development of the SanQoL outcome measure as: (1) participants in the qualitative research informing attribute identification[9] and (2) participants in the piloting and cognitive interviews informing item development.[28]

## RESULTS

We sampled individuals from 424 different households across 272 compounds (clusters), of which 130 were control and 142 intervention, during April–May 2019 (figure 1). In some compounds, only one man or woman was eligible (mean respondents per cluster: 1.6). The response rate among eligible participants was 99%. There was no evidence of difference in background characteristics between intervention/control at the 5% level, except for the wealth index score (table 2). There were two further differences at the 10% level (age and secondary education). People living in intervention compounds were slightly wealthier than controls, but assets that were different were the less expensive ones (eg, watch, radio).

## Primary outcome

The intervention was associated with an adjusted gain in SanQoL of 0.34 (95% CI 0.29 to 0.38), noting that SanQoL is measured on a 0–1 scale and the control mean was 0.49 (table 3). Full regression results are in online supplemental appendix E. The effect size was very large at 1.6 SD. None of the three covariates (wealth, gender and aged 60+) were significant at the 5% level. The additional exploratory analyses regressing on each of the five SanQoL attributes individually are reported in online supplemental appendix E.

## Secondary outcomes

Measured on the sanitation VAS, which is scored 0–10, the intervention was associated with a 2.9 point gain (95% CI 2.4 to 3.4) (table 3). The effect size was very large at 1.3 SD,

similar to that for SanQoL. Considering WHO-5 mental well-being, measured on a 0–100 scale, the intervention was associated with a 6.2 point gain (95% CI 0.3 to 12.2) (table 3). The effect size was small at 0.2 SD. There was evidence for people aged 60+ having lower mental well-being outcomes across the sample (online supplemental appendix E). For neither secondary outcome were any other covariates significant at the 5% level.

## Moderating effects

Recalling that our study was not powered for subgroup analyses, including an interaction term for the intervention with gender provided no evidence for any outcome that women benefited more from better toilets than men (table 4). There was also no evidence that people aged 60+ benefitted more than under-60s, for any outcome.

## Sensitivity analyses

When only covariates significantly different between groups at the 10% level were included (table 2), there was no meaningful difference to results for any of the three outcomes (online supplemental appendix F). Second, when further covariates hypothesised as predicting SanQoL and VAS were included, there was no evidence of omitted variable bias in terms of the sizes and p values of coefficients on the intervention variable. However, the coefficient on the binary covariate for sharing the toilet was negative and significant at the 1% level in both SanQoL and VAS regressions. This finding, indicating that those across the sample sharing toilets with other households had worse SanQoL, is explored as a factorial interaction in online supplemental appendix G. Third, when all covariates hypothesised as predicting mental well-being were included in the WHO-5 regression, there was no evidence of omitted variable bias. Fourth, using a GEE or OLS specification did not affect headline results for SanQoL or VAS. The effect on WHO-5 was significant only at the 10% level in the OLS regression, but OLS is unlikely to be appropriate for the hierarchical structure of our data. Furthermore, residuals were bimodally

| | Means | | Unadjusted models | | Adjusted models | | |
|---|---|---|---|---|---|---|---|
| **Outcome** | **Control (n=202) Mean (SE)** | **Interv'n (n=222) Mean (SE)** | **Unadjusted difference (95% CI)** | **P value** | **Adjusted difference (95% CI)** | **P value** | **Adjusted effect size (Cohen's d)** |
| SanQoL (0–1 scale) | 0.49 (0.02) | 0.83 (0.01) | 0.34*** (0.29 to 0.38) | <0.001 | 0.34*** (0.29 to 0.39) | <0.001 | 1.6 |
| Sanitation VAS (0–10 scale) | 4.1 (0.2) | 7.0 (0.1) | 2.9*** (2.4 to 3.4) | <0.001 | 2.9*** (2.4 to 3.4) | <0.001 | 1.3 |
| WHO-5 (0–100 scale) | 54.4 (1.9) | 58.7 (1.9) | 5.6* (-0.4 to 11.6) | 0.065 | 6.2** (0.3 to 12.2) | 0.041 | 0.2 |

**Table 3** Effects on primary and secondary outcomes

Adjusted models include gender, aged 60+ and wealth score as covariates. SEs are clustered at the compound level. *, **, *** indicate significance at the 10%, 5% and 1% level. Detailed regression output is in online supplemental appendix E.
SanQoL, Sanitation-related Quality of Life; VAS, Visual Analogue Scale.

**Table 4** Moderating effects on outcomes by gender and aged 60+

| Outcome | Gender interaction model | | | | Age interaction model | | | |
|---|---|---|---|---|---|---|---|---|
| | Female | | Female * intervention | | Aged 60+ | | Aged 60+ * intervention | |
| | coeff. | P value | coeff. | P value | coeff. | P value | coeff. | P value |
| SanQoL (0–1 scale) | −0.02 | 0.49 | 0.03 | 0.49 | −0.03 | 0.57 | 0.03 | 0.62 |
| Sanitation VAS (0–10 scale) | −0.45 | 0.06* | 0.29 | 0.37 | −0.14 | 0.75 | −0.02 | 0.98 |
| WHO-5 (0–100 scale) | −2.91 | 0.25 | −0.77 | 0.84 | −10.6 | 0.01** | −4.09 | 0.47 |

Interaction models includes gender, aged 60+, and wealth score as covariates, in addition to the interaction term indicated in columns. SEs are clustered at the compound level. *, **, *** indicate significance at the 10%, 5% and 1% level. Detailed regression outputs are in online supplemental appendix E. Coeff.=coefficient.

distributed in the WHO-5 OLS regression, suggesting GLMM/GEE specifications are preferred. We conclude from our sensitivity analyses that our main models were appropriate for testing our hypotheses. Comparing effects between the two intervention toilet designs, there was weak evidence (p=0.079) for users of STs having higher SanQoL than users of CSBs (online supplemental file 1 - Appendix E).

## DISCUSSION

In this observational study building on the design of the earlier MapSan trial, we find that users of high-quality shared toilets experienced a 1.6 SD gain in SanQoL compared with pit latrines, and a 0.2 SD gain in broader mental well-being. A non-randomised controlled trial of this intervention found no effect on under-5 health outcomes.[27] Therefore, our findings demonstrate that better toilets can improve people's lives beyond infectious disease, at a time when several randomised trials have questioned the health effects of sanitation improvements.[20]

Since all people in intervention compounds were previously sharing a low-quality toilet with the same people in the same location, the mechanism driving our results is likely to be the specific characteristics of intervention toilets. Solid walls and doors likely improved perceptions of privacy, safety and shame compared with PLs with makeshift walls and doors (photos in online supplemental appendix A). The pour-flush interface was likely to have reduced smells and visible faeces compared with PLs without water seals, improving perceptions of disgust, shame and health risk. Users value such toilet characteristics—a choice experiment in urban Zambia found willingness to pay (WTP) additional rent for solid toilet doors was 8% of median monthly rent, and WTP for flush toilets as opposed to pit latrines was 5% of rent.[40]

While it is intuitive that people using better-quality toilets experience more privacy or less disgust, our contribution is in quantifying this to inform decisions based on comparative effectiveness, which has not previously been done.[41] The fact that SanQoL is specific to sanitation limits its broader relevance. However, such 'condition-specific'

outcomes focused on experienced symptoms (eg, of arthritis or asthma) within only a few QoL domains are regularly used to evaluate interventions targeting those specific problems.[42] The small effect on mental well-being was expected, as it is a more distal outcome than SanQoL. A previous cross-sectional study identified associations of urban sanitation access with WHO-5,[21] and our contribution is in evaluating a specific urban sanitation intervention.

Despite willingness to contribute financially to 10%–15% of capital costs being an enrolment criterion for both intervention and control in MapSan, it is possible that wealthier people were more likely to uptake the intervention due to being able to afford this contribution. Since our survey was 4 years after the intervention, wealth differences could be as a result of the intervention. However, any wealth effect might be in the other direction since intervention households reported spending substantially more than controls on both cleaning and maintenance.[43]

Our hypotheses about women benefitting more than men and elderly people more than non-elderly were not supported. While our study was not powered for these analyses, p values on interaction terms were very large in all cases (table 4), suggesting that increased power may not have altered results. In the main analyses without interactions (table 3), neither gender nor aged 60+ covariates were significant at the 5% level, except in the case of aged 60+ for the mental well-being outcome (online supplemental appendix E). These hypotheses were informed by the qualitative literature,[4 10–13] and we are not aware of any quantitative evidence for sanitation interventions disproportionately benefitting women or older people for any QoL outcome. Evidence for gendered monetary valuation of toilet attributes in the WTP literature is also mixed. Studies of WTP for latrine slabs (in Tanzania),[44] and for other toilet attributes (in Zambia)[39 44] find no evidence of gendered differences in valuations. A WTP study in Kenya found higher uptake of discount vouchers among men.[44]

Limitations of our study include that we relied on the controlled before-and-after design of a previous trial in which the intervention was not randomly allocated, risking

selection bias. Our eligibility criterion of having lived on the compound since before the intervention aimed to reduce risk of bias from in-migration to intervention compounds as a result of the high-quality sanitation facilities. This criterion may have introduced bias if differential rates of out-migration took place, though the MapSan trial found no evidence of this between 2015 and 2018.[27] Our eligibility criterion of still using the type of toilet consistent with intervention/control status aimed to ensure the integrity of validity assessments reported elsewhere.[28] However, it also meant that this study does not provide an unbiased estimate of the effect of the intervention. Our design necessitates adjusting for covariates which may be imprecisely measured, and the absence of pre-intervention SanQoL data precluded adjustment for baseline values. While our comparison groups were well-balanced overall, and we adjusted for unbalanced covariates, we cannot account for unobserved confounding. The magnitude of the effect size (1.6 SD) for our primary outcome means it is unlikely to be explained by bias alone. The finding for the mental well-being outcome is more precarious, however, due to its smaller effect size (0.2 SD) and higher p value (0.04).

As with any subjective self-reported outcome, there is risk of reporting bias which is difficult to account for, though we assume that any measurement error was not correlated with toilet type. In introducing themselves, fieldworkers emphasised that they were not linked to the implementing NGO or government, but intervention respondents may have wanted to appear grateful and control respondents may have wanted to appear badly off. Reporting bias could pose more of a risk to the mental well-being finding with its higher p value, but the WHO-5 questions do not refer to sanitation in any way, so may have been perceived by respondents as being less linked to the intervention. A final limitation is that we did not prespecify a statistical analysis plan.

Since physical environments and sanitary conditions in these parts of Maputo are similar to large portions of other Mozambican cities, it is likely that findings could be generalisable to those settings, as well as parts of other cities in many African countries. However, social environments differ within and beyond Mozambique, and are likely to influence the relationships explored. Future intervention trials should include QoL outcomes, since these benefits are highly valued by users. Changes in sanitation-specific QoL outcomes such as privacy and disgust are likely to suffer from fewer confounding factors than infectious disease outcomes, since they are more proximal to the service being provided.

## CONCLUSION

QoL outcomes are valued by toilet users and can be improved by sanitation interventions. If applied in future impact evaluations alongside health outcomes, SanQoL, WHO-5 and similar measures could help sanitation decision-makers understand which types of sanitation interventions most improve people's QoL as well as prevent disease. Some interventions may improve one but not the other. QoL indices with weighting derived from respondent valuation tasks, such as SanQoL, can also be used in economic evaluation to identify interventions which are most efficient use of resources, not only those which are most effective.

**Author affiliations**
[1]Department of Global Health and Development, London School of Hygiene & Tropical Medicine, London, UK
[2]Department of Disease Control, London School of Hygiene & Tropical Medicine, London, UK
[3]WE Consult, Maputo, Mozambique
[4]Instituto Nacional de Saúde, Maputo, Mozambique
[5]Department of Environmental Science and Engineering, University of North Carolina, Chapel Hill, North Carolina, USA
[6]Department of Medical Statistics, London School of Hygiene and Tropical Medicine, London, UK
[7]Nuffield Department of Population Health, University of Oxford, Oxford, Oxfordshire, UK

**Acknowledgements** We greatly appreciated the cooperation of survey respondents. We also appreciated the efforts of the fieldworkers employed by WE Consult: Euclimia Titosse, Carla Panguene, João-Pedro Guambe and Faustino Benzane. The authors gratefully acknowledge the technical and logistical support received from Vasco Parente and his team at WSUP Mozambique. We benefitted from the comments of Roxanne Kovacs and seminar participants at the Global Health Economics Centre at LSHTM, as well as from Britta Augsburg and Antonella Bancalari.

**Contributors** IR conceptualised the study and developed the methods as part of a PhD, with support from OC/GG/CO. ZA and IR refined the methods through discussion, fieldworker training, cognitive interviews and piloting. ZA led the fieldwork team and curated the data. IR analysed the data and wrote the original draft. All authors inputted into subsequent drafts. OC/RN/JB secured the funding. IR is the guarantor.

**Funding** This work was supported by the Economic and Social Research Council through a PhD studentship. The fieldwork was funded by the Bill & Melinda Gates Foundation (OPP1137224).

**Disclaimer** The funders had no role in the identification, design, conduct, or reporting of the analysis.

**Map disclaimer** The inclusion of any map (including the depiction of any boundaries therein), or of any geographic or locational reference, does not imply the expression of any opinion whatsoever on the part of BMJ concerning the legal status of any country, territory, jurisdiction or area or of its authorities. Any such expression remains solely that of the relevant source and is not endorsed by BMJ. Maps are provided without any warranty of any kind, either express or implied.

**Competing interests** None declared.

**Patient and public involvement** Patients and/or the public were not involved in the design, or conduct, or reporting, or dissemination plans of this research.

**Patient consent for publication** Not applicable.

**Ethics approval** The study received prior approval from the Comité Nacional de Bioética para a Saúde (IRB00002657) at the Ministry of Health in Mozambique, and the Research Ethics Committee of the London School of Hygiene and Tropical Medicine (Ref: 14609). Informed, written consent was obtained from all participants. Participants gave informed consent to participate in the study before taking part.

**Provenance and peer review** Not commissioned; externally peer reviewed.

**Data availability statement** Data are available in a public, open access repository. Deidentified individual participant data, data dictionary and replication code are available open access on the LSHTM data repository (ref. 39).

**ORCID iD**
Ian Ross http://orcid.org/0000-0002-2218-5400

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
