## [Reviewer comments · BMJ Open]

ARTICLE DETAILS

TITLE (PROVISIONAL)	Impact of a sanitation intervention on quality of life and mental wellbeing in low-income urban neighbourhoods of Maputo, Mozambique: an observational study
AUTHORS	Ross, Ian; Greco, Giulia; Adriano, Zaida; Nala, Rassul; Brown, Joe; Opondo, Charles; Cumming, Oliver

VERSION 1 – REVIEW

REVIEWER	Hennegan, Julie Johns Hopkins University Bloomberg School of Public Health, Environmental Health and Engineering
REVIEW RETURNED	29-Mar-2022

GENERAL COMMENTS	This paper uses a cross-sectional survey undertaken with communities who had previously participated in a controlled before-after study to investigate ongoing differences in sanitation-related quality of life and wellbeing. It is a valuable paper highlighting the potential gains in QoL that can be achieved through investment in improved sanitation. My main concerns with the methods and analyses are (1) it is unclear if analyses were exploratory or followed an a-priori protocol, and if not, how analysis decisions were made. (2) Authors have not justified why elderly age and gender are adjusted for in main analyses and then pursued as moderators/sub-groups. I do not think including these as covariates in adjusted models is justified. We would hypothesise that the effect of having a better toilet may be larger or smaller across these sub-groups but not that they confound the association between the intervention (having a better toilet) and the outcome. I also have some more minor concerns about the sampling restrictions which requires more attention in the discussion. The paper is very difficult to follow needs significant revision prior to publication to ensure readers can follow and understand the results (and limitations) of the work. **Title and abstract** I found the title confusing. It describes testing the 'impact' of the intervention, but then states it is an observational study and doesn't actually specify the study design. I think at minimum this needs to specify a cross-sectional survey rather than 'an observational' study. I was also confused by the use of 'interview' in the abstract. This was a survey. *Introduction*
---

I found the introduction difficult to follow, we seem to jump from the need to consider QoL, to what QoL is, to measuring QoL – all in the same paragraph. Hypotheses don't appear until the analysis section of the methods, but these should come earlier in the paper and flow (and be justified) from the literature review in the introduction. Many research questions (hypotheses) are not foregrounded in the introduction at all (e.g., gender and age differences, investigation of differences across QoL dimensions and why this is included in the study)

A few specific notes:

Page 4, lines 18-19, I found this sentence difficult it might be helpful for readers to define your terms or rephrase.

Lines 28-29 what QoL outcomes? This seems like it should be in the methods. You provide three examples above, then mention seven, and its not clear what "each" is referring to.

Lines 36-39 - I believe that systematic review was largely of qualitative research which is important to note in the context of this sentence.

****Methods****

The study design section describes the enrolment of participants into the MapSan trial. This outlines that the control group was selected to match the intervention compounds on criteria 1-4 (criteria 3 being that inhabitants are willing to contribute financially to constructing the trial sanitation upgrades). Authors then state that for this study they sought to reduce selection bias by selecting participants that were still using the intervention facility, and for the control had not improved their sanitation. This approach seems to create 'ideal' or more extreme versions of the intervention and control (with successful and long-lasting use of the new sanitation) or prolonged unimprovement in the sanitation). Given criteria 3, we may well expect control groups to have upgraded their sanitation independently, and we could expect new selection bias by excluding those who then upgraded after the trial. I would presume that the trial followed an intention to treat analysis and did not restrict outcome assessment to only those who received the intervention per-protocol. The sample here is something akin to per-protocol ++ where you are only including those who received the intended intervention for an even longer period of time than originally designed, and allow no improvement in the control. While the authors have labelled this as an observational study, some of the conclusions are still worded more similarly to a trial, but without being held accountable to the biases in analysis we would interrogate if this was a trial. This limitation should be noted in the discussion and in doing revisions authors should closely review the language used to note that this tests the differences in QoL between participants with very different sanitation infrastructure, but does not serve to provide an unbiased effect estimate of the intervention.

Page 7 - It is not appropriate to have core hypotheses that impact on the statistical analysis for the study outlined only in appendices. It makes the paper laborious to read (and review) and means that readers are going to miss key information for interpreting the results of the study. The content from Appendix B (table B-1) needs to be integrated into the main manuscript. This kind of information should be integrated into the introduction to justify the study approach and hypotheses outlined as they impact analysis. The hypotheses are quite difficult for a reader to follow (e.g., page 7 line 38-40 "with 1b and 1c the same for VAS and WHO-5"). This

is both colloquial and expecting a lot from readers. Suggest revision to simplify. There also seem to be multiple numbering systems (i) hypothesis 1? This then flows on to some difficult wording in the analysis section, Page 8, lines 35-37 we jump from Hypothesis 1b 1c to hypothesis 3, then back to hypothesis 2a where we specify the 'same models'...

Hypothesis 3 (or III) is unclear – 'some SanQoL attributes contribute more to overall differences than others'. Is this suggesting that improvements in SanQoL mediate improvements in VAS and WHO-5? Or suggesting that changes on some of the individual SanQoL items had larger changes than other (e.g., privacy rather than safety), in which case contribute to what differences? The overall score? It doesn't make sense to word this as 'contribute to', more that changes were greater on some domains. This hypothesis (research question) needs to be justified.

I am confused by the analysis section paragraphs starting page 8 line 22 and then page 8 line 35. The first paragraph here seems to suggest that authors adjusted for gender and being elderly (despite hypothesising a moderating effect), but then later down in the next paragraph authors state they specified an interaction effect for these variables (consistent with the moderation hypothesis).

In testing the main effect of the intervention, why adjust for gender and age as if they were confounders? Hypotheses clearly outline these as moderating effects. This seems like this was re-introduced again in the sensitivity analyses.

****Results****

Again, as a reader I am struggling to follow the threads of the different hypotheses, results jump from one to another, to sensitivity analyses and back again, with few sub-headings to orientate the reader.

As noted above, adjustment for gender and elderly age are inappropriate.

I find it odd that the authors have included in the main text of the paper the demographic comparisons between the intervention and control, but then moved analyses related to primary hypotheses (moderating effects of gender and age on QoL) to the appendix. This feels curiously like dismissing hypotheses that were not supported. I also find it odd that this is listed as a primary hypothesis but the study wasn't powered to test these moderating effects. Was there an a-priori analytic protocol that was followed? The investigation of variation across different SanQoL attributes its not clearly set up in the introduction. It comes in as a bit of a surprise and its not clear why this was prioritized. Was this outlined a-priori? Or was this pursued as an exploratory analysis when sub-group differences weren't identified for elderly age and gender.

Discussion

The limitations noted above should be discussed in the discussion. I find it concerning that the discussion omits any discussion about the unsupported gender and age hypotheses and focuses only on the supported hypotheses. The discussion should fairly address all a-priori specified research questions, along with alternative explanations.

The conclusion seems to pivot to the importance of QoL in contrast to health in trials, this wasn't the focus of your study and extends well beyond the results presented. It is also plausible that

	the interventions that improve QoL are not the same as those with greater benefits to public health and a whole host of other considerations that might need to be addressed in some of these statements. I suggest revising to summarize the implications for policy and practice of YOUR findings.
--	--

REVIEWER	SSemugabo, Charles Makerere University CHS, Department of Disease Control and Environmental Health
REVIEW RETURNED	09-Apr-2022

GENERAL COMMENTS	The authors need to add a paragraph that contextualise the Sanitation situation within Maputo City and Mozambique at large in the introduction. In the study site description, the authors do not highlight access to water in this community yet it's a key requirement for the pour-flush toilet technology. The authors should clearly explain how they determined the sample size for the study. What is the 398? Is it compounds or individual participants. What was the study unit? And how did this impact their sample size calculation. All these aspects are not clear. How did the authors ensure that participants in the control group were still using the same sanitary facility at the endline evaluation? What was the criteria for selecting control and intervention compounds? The authors should clearly explain how the randomization was done.
--

VERSION 1 – AUTHOR RESPONSE

	Reviewer 1	Response	Action taken
	Reviewer's summary		
1	"My main concerns with the methods and analyses are (1) it is unclear if analyses were exploratory or followed an a-priori protocol, and if not, how analysis decisions were made. (2) Authors have not justified why elderly age and gender are adjusted for in main analyses and then pursued as moderators/sub-groups. I do not think including these as covariates in adjusted models is justified. We would hypothesise that the effect of having a better toilet may be larger or smaller across these sub-groups but not that they confound the association between the intervention (having a better toilet) and the outcome. I also have some more minor concerns about the sampling restrictions which requires more attention in the discussion. The	We thank the reviewer for their comments. These summary points are addressed below where the reviewer has outlined them in more detail.	Outlined below.

	Reviewer 1	Response	Action taken
	paper is very difficult to follow needs significant revision prior to publication to ensure readers can follow and understand the results (and limitations) of the work.		
	Title and abstract		
2	I found the title confusing. It describes testing the 'impact' of the intervention, but then states it is an observational study and doesn't actually specify the study design. I think at minimum this needs to specify a cross-sectional survey rather than 'an observational' study.	This manuscript is not reporting a cross-sectional study in the usual sense of that term, because it is a comparison of participants from two trial arms. Cochrane define an observational study as “A study in which the investigators do not seek to intervene, and simply observe the course of events. Changes or differences in one characteristic (e.g. whether or not people received the intervention of interest) are studied in relation to changes or differences in other characteristic(s) (e.g. whether or not they died), without action by the investigator.” https://epoc.cochrane.org/sites/epoc.cochrane.org/files/public/uploads/SUR-E-Guides-v2.1/Collectedfiles/source/glossary.html We did struggle to describe the study design succinctly for the title, and calling it “an observational study” seemed the most straightforward and brief way to do so (see e.g. https://doi.org/10.1016/S2468-2667(20)30090-6). It is common for observational studies to be reported as assessing “impact” of interventions (e.g. see link above, and also https://doi.org/10.1016/S0140-6736(17)31119-4 and https://doi.org/10.1016/j.lana.2021.100128).	n/a

	Reviewer 1	Response	Action taken
		Therefore, we feel that calling this an observational study to assess impact is justified. We agree with the reviewer though that the limitations of this study design should have been better explained, which we have now done in response to the separate helpful comments from the reviewer (below).	
3	I was also confused by the use of 'interview' in the abstract. This was a survey.	We agree "survey" is more appropriate.	Amended to "surveyed" (p. 2 line 7):
	Introduction		
4	I found the introduction difficult to follow, we seem to jump from the need to consider QoL, to what QoL is, to measuring QoL – all in the same paragraph.	We have substantially rewritten the introduction. The logic of the 4 paragraphs is now:  1. Sanitation affects outcomes beyond infectious disease, which are part of QoL. 2. Individual and environmental factors may moderate the effect of sanitation interventions on QoL 3. QoL outcomes are important to people, so they probably influence uptake of sanitation interventions and are thus important to measure. 4. However, only a few studies have measured sanitation-related QoL outcomes, and none in the context of an urban sanitation intervention. 	Revised whole introduction.
5	Hypotheses don't appear until the analysis section of the methods, but these should come earlier in the paper and flow (and be justified) from the literature review in the introduction. Many research questions (hypotheses) are not foregrounded in the introduction at all (e.g., gender and age differences, investigation of differences across QoL dimensions and why this is included in the study)	We have added a paragraph explaining the rationale for gender and age being important, and have foregrounded the moderation analyses in methods and results as well (see below). To make the paper more focused, we have moved the analysis of SanQoL attributes to the annex, so do not	Added paragraph (p. 4 line 13ff) "Many factors are hypothesised to moderate effects of sanitation interventions on QoL outcomes. ^{5,9} Some of these factors relate to individual characteristics. For example, women and girls might be at

	Reviewer 1	Response	Action taken
		propose to emphasise this in the introduction.	greater risk of infringements to their sanitation-related safety and privacy than men and boys. ^{4,10,11} People with reduced mobility such as older or disabled people may be more likely to fear falling into a pit latrine. ^{12,13} Other factors relate to the environment. For example, someone using a nearby toilet in an urban neighbourhood perceived as unsafe may feel less secure using it at night than someone in a rural area. ¹⁴
6	A few specific notes: Page 4, lines 18-19, I found this sentence difficult it might be helpful for readers to define your terms or rephrase.	We have deleted this sentence.	n/a
7	Lines 28-29 what QoL outcomes? This seems like it should be in the methods. You provide three examples above, then mention seven, and its not clear what "each" is referring to.	We have reframed the introduction, so this phrasing is no longer there.	n/a
8	Lines 36-39 - I believe that systematic review was largely of qualitative research which is important to note in the context of this sentence.	We agree, and have added that point.	Added "qualitative" (p. 4 line 30)
	Methods		
9	The study design section describes the enrolment of participants into the MapSan trial. This outlines that the control group was selected to match the intervention compounds on criteria 1-4 (criteria 3 being that inhabitants are willing to contribute financially to constructing the trial sanitation upgrades). Authors then state that for this study they sought to reduce selection bias by selecting participants that were still using the intervention facility, and for the control had not improved their sanitation. This approach seems to create 'ideal' or more extreme versions of the	The alternative to our chosen study design would have been to sample participants who lived on intervention/control compounds regardless of when they started living there or what sanitation facility they now used. Two limitations of this alternative study design caused us to settle on our chosen design. First, given migration, the alternative study design would have meant sampling people who moved onto to the compound after the intervention, possibly	Added further rationale for criteria to methods (p.6, line 2) "This criterion aimed to ensure a sufficient sample of people using low-quality toilets for the purposes of validity assessments reported elsewhere,²⁸ in the context of unknown levels of subsequent intervention and upgrading."

	Reviewer 1	Response	Action taken
	intervention and control (with successful and long-lasting use of the new sanitation) or prolonged unimprovement in the sanitation). Given criteria 3, we may well expect control groups to have upgraded their sanitation independently, and we could expect new selection bias by excluding those who then upgraded after the trial. I would presume that the trial followed an intention to treat analysis and did not restrict outcome assessment to only those who received the intervention per-protocol. The sample here is something akin to per-protocol ++ where you are only including those who received the intended intervention for an even longer period of time than originally designed, and allow no improvement in the control. While the authors have labelled this as an observational study, some of the conclusions are still worded more similarly to a trial, but without being held accountable to the biases in analysis we would interrogate if this was a trial. This limitation should be noted in the discussion and in doing revisions authors should closely review the language used to note that this tests the differences in QoL between participants with very different sanitation infrastructure, but does not serve to provide an unbiased effect estimate of the intervention.	attracted by the better sanitation facilities. Second, only one survey to achieve multiple objectives was possible within the available funding. A second objective of the survey was to explore the validity/reliability of the SanQoL measure (study reported elsewhere https://doi.org/10.1002/hec.4462). If we had included compounds which had autonomously upgraded or subsequently received an NGO intervention, this would have reduced the sample of people using lower quality toilets. In turn, this would have reduced the power of hypothesis tests in the construct validity assessment. At the time of study design, we did not know the extent of autonomous upgrading and subsequent intervention, which ultimately was 18% (29/163 – see Figure 1) of the compounds we approached. We were worried it could be higher than this and harm the validity assessment. This latter point has been added to the manuscript. We have expanded the limitations paragraph of the discussion. We have also explained that our study does not provide an unbiased effect estimate of the intervention, and reduced the conclusions section. Our hypotheses section and results section were already worded as associations and remain so. However, we are estimating effect/impact (see response to comment #2 above). On the point about “received the intended intervention for an even	Expanded relevant part of limitations paragraph (p.14, line 6). “Our eligibility criterion of having lived on the compound since before the intervention aimed to reduce risk of bias from in-migration to intervention compounds as a result of the high-quality sanitation facilities. This criterion may have introduced bias if differential rates of out-migration took place, though the MapSan trial found no evidence of this between 2015-2018.²⁷ Our eligibility criterion of still using the type of toilet consistent with intervention/control status aimed to ensure the integrity of validity assessments reported elsewhere.²⁸ However, it also meant that the present study does not provide an unbiased estimate of the effect of the intervention.”

	Reviewer 1	Response	Action taken
		longer period of time than originally designed”, we might expect the effect to be in the other direction. With a toilet with a useful life of 15 years, a slight decline in toilet quality (and therefore smaller QoL effects over time) might be expected between years 0 and 4. So we might expect to have observed a weaker effect on outcomes than if they had been measured earlier, e.g. 1 year after delivery.	
10	Page 7 - It is not appropriate to have core hypotheses that impact on the statistical analysis for the study outlined only in appendices. It makes the paper laborious to read (and review) and means that readers are going to miss key information for interpreting the results of the study. The content from Appendix B (table B-1) needs to be integrated into the main manuscript. This kind of information should be integrated into the introduction to justify the study approach and hypotheses outlined as they impact analysis.	This content was annexed for word count reasons, but we have integrated it back into the manuscript, and addressed in the introduction.	Put hypotheses in specific section and expanded (p.7 line 21ff)
11	The hypotheses are quite difficult for a reader to follow (e.g., page 7 line 38-40 “with 1b and 1c the same for VAS and WHO-5”). This is both colloquial and expecting a lot from readers. Suggest revision to simplify. There also seem to be multiple numbering systems (i) hypothesis 1? This then flows on to some difficult wording in the analysis section, Page 8, lines 35-37 we jump from Hypothesis 1b 1c to hypothesis 3, then back to hypothesis 2a where we specify the ‘same models’...	We have simplified the presentation of hypotheses and removed numbering.	Adjusted throughout methods and results.
12	Hypothesis 3 (or III) is unclear – ‘some SanQoL attributes contribute more to overall differences than others’. Is this suggesting that improvements in SanQoL mediate improvements in VAS and WHO-5? Or suggesting that changes on some of the individual	We agree with the reviewer that this could have been more clearly presented. To focus the paper more, we now introduce this content as additional analysis not a core hypothesis, and	Edited as indicated (p.9, line 2) “As an additional exploratory analysis, we assessed effects on each of the five SanQoL attributes individually, by

	Reviewer 1	Response	Action taken
	SanQoL items had larger changes than other (e.g., privacy rather than safety), in which case contribute to what differences? The overall score? It doesn't make sense to word this as 'contribute to', more that changes were greater on some domains. This hypothesis (research question) needs to be justified.	shifted the associated results to the annex.	regressing on their raw scores (ranging 0-3). The rationale was to explore whether larger effect sizes were seen on some dimensions than others." (p.11, line 6) "The additional exploratory analyses regressing on each of the five SanQoL attributes individually are reported in Online Appendix E."
13	I am confused by the analysis section paragraphs starting page 8 line 22 and then page 8 line 35. The first paragraph here seems to suggest that authors adjusted for gender and being elderly (despite hypothesising a moderating effect), but then later down in the next paragraph authors state they specified an interaction effect for these variables (consistent with the moderation hypothesis). In testing the main effect of the intervention, why adjust for gender and age as if they were confounders? Hypotheses clearly outline these as moderating effects. This seems like this was re-introduced again in the sensitivity analyses.	We should have been clearer that the reason for adjusting for gender and being over-60 in the main models. It was not that they are hypothesised as being confounders, which would imply an expected association with the exposure (the intervention, in this case). Rather, they are considered as predictive of the participant's response to the intervention. In analysis of RCTs, for example, it is common that variables considered predictive of the outcome are included in the analysis. ^{9,10} While this is an observational study, a similar principle applies. In other words, since the outcome is expected to vary by age or gender (per our moderation hypotheses), that is a good reason to adjust. Fitting an interaction involving these variables simply relaxes one assumption of adjustment: instead of assuming that the effect of the intervention is the same across age-groups (adjustment), fitting an interaction allows the effect to vary across age-groups. Therefore, we do	Edited methods: (p.8, line 34) "Second, we included binary variables for gender and being elderly (aged 60+), because they are considered predictive of the participant's response to the intervention (as hypothesised in moderation analyses outlined above). ^{35,36} "

	Reviewer 1	Response	Action taken
		not believe that adjusting for these covariates violates any principle. We have updated the methods section to explain the rationale for adjustment. If readers are interested in results without these adjustments, the first sensitivity analysis includes only covariates significantly different between groups at the 10% level (i.e. gender and aged 60+ are not adjusted for).	
	Results		
14	Again, as a reader I am struggling to follow the threads of the different hypotheses, results jump from one to another, to sensitivity analyses and back again, with few sub-headings to orientate the reader.	We agree this could have been clearer, and have added sub-headings to the results section.	Added sub-headings to the results section.
15	As noted above, adjustment for gender and elderly age are inappropriate. I find it odd that the authors have included in the main text of the paper the demographic comparisons between the intervention and control, but then moved analyses related to primary hypotheses (moderating effects of gender and age on QoL) to the appendix. This feels curiously like dismissing hypotheses that were not supported. I also find it odd that this is listed as a primary hypothesis but the study wasn't powered to test these moderating effects. Was there an a-priori analytic protocol that was followed?	On adjustment, see above. We have moved hypotheses to a specific sub-section and separated out paragraphs on the main hypotheses versus the additional moderation analyses. We have added a line to methods that analyses were not pre-specified, and also added the same as a limitation. We agree with the reviewer that the moderation analyses are important. In earlier versions they had been in the manuscript, but were pushed to the annex due to BMJ Open permitting only five figures and tables. Since we have instead moved the attribute-level exploration to the annex (see above), this freed up space for the moderation	Hypotheses section edited as indicated above. Pre-specification amends: (p. 8 line 14) "Analyses were not pre-specified." (p. 14 line 28) "A final limitation is that we did not pre-specify a statistical analysis plan." Moderation analyses presented in new Table 4 (p. 12)

	Reviewer 1	Response	Action taken
		analyses to be restored to the main manuscript.	
16	The investigation of variation across different SanQoL attributes its not clearly set up in the introduction. It comes in as a bit of a surprise and its not clear why this was prioritized. Was this outlined a-priori? Or was this pursued as an exploratory analysis when sub-group differences weren't identified for elderly age and gender.	We have edited the methods section to clarify that these were additional analyses, to explore whether larger effect sizes were seen on some dimensions than others. We do not propose to over-burden the introduction, since specific hypotheses are not proposed for this analysis. As above, we have added a line to methods stating that analyses were not pre-specified.	Amends to methods (p.9 line 2) "As an additional exploratory analysis, we assessed effects on each of the five SanQoL attributes individually, by regressing on their raw scores (ranging 0-3). The rationale was to explore whether larger effect sizes were seen on some dimensions than others."
	Discussion		
17	The limitations noted above should be discussed in the discussion. I find it concerning that the discussion omits any discussion about the unsupported gender and age hypotheses and focuses only on the supported hypotheses. The discussion should fairly address all a-priori specified research questions, along with alternative explanations.	We have added limitations related to the eligibility criteria, as mentioned in response to comment #9 above. On the moderation hypotheses, this content was removed to meet word limits and has been reinstated.	Limitations about eligibility added (see response to comment #9 above). Paragraph added discussing moderation results. (p.13, line 32ff) "Our hypotheses about women benefitting more than men and elderly people more than non-elderly were not supported. While our study was not powered for these analyses, p-values on interaction terms were very large in all cases (Table 4), suggesting that increased power may not have altered results. In the main analyses without interactions (Table 3), neither gender nor aged 60+ covariates were significant at the 5% level, except in the case of aged 60+ for the mental wellbeing outcome (Online Appendix E). These hypotheses were informed by

	Reviewer 1	Response	Action taken
			the qualitative literature,^{4,10–13} and we are not aware of any quantitative evidence for sanitation interventions disproportionately benefitting women or older people for any QoL outcome. Evidence for gendered monetary valuation of toilet attributes in the willingness to pay literature is also mixed. Studies of WTP for latrine slabs (in Tanzania),⁴³ and for other toilet attributes (in Zambia)^{39,44} find no evidence of gendered differences in valuations. A WTP study in Kenya found higher uptake of discount vouchers amongst men.⁴³”
18	The conclusion seems to pivot to the importance of QoL in contrast to health in trials, this wasn't the focus of your study and extends well beyond the results presented. It is also plausible that the interventions that improve QoL are not the same as those with greater benefits to public health and a whole host of other considerations that might need to be addressed in some of these statements. I suggest revising to summarize the implications for policy and practice of YOUR findings.	The results in this manuscript show an effect on QoL. The results of the MapSan trial itself show no effect on child health.¹⁵ So we think the points we make in the conclusion are in line with the results of the trial. It is plausible that other recent sanitation trials with null health effects may have seen QoL effects if they had measured them. However, to make the conclusion more succinct and focused on the above point, we have removed the later content about QoL outcomes being more quickly/easily measured. The reviewer's point about “interventions that improve QoL are not the same as those with greater benefits to public health” is a useful one and we have added this to the conclusion.	Amended sentences to read (p. 15 line 4): “If applied in future impact evaluations alongside health outcomes, SanQoL, WHO-5 and similar measures could help sanitation decision-makers understand which types of sanitation interventions most improve people's QoL as well as prevent disease. Some interventions may improve one but not the other.” Removed below sentences: “The likelihood of positive health externalities remains a core rationale for public investment in sanitation. However, in the almost certain absence of intervention-specific data on health effects, policy-makers are likely to be willing to make more informed decisions on

	Reviewer 1	Response	Action taken
			the basis of QoL outcomes which are more easily and quickly measured in their specific setting.”

	Reviewer 2	Response	Action taken
1	The authors need to add a paragraph that contextualise the Sanitation situation within Maputo City and Mozambique at large in the introduction.	We thank the reviewer for their time and effort to review our manuscript. We had a description of the Maputo sanitation context in the “setting” sub-section under methods, but have added a point about Mozambique more broadly, in line with the reviewer’s recommendation.	Added sentence (p. 5 line 4): “In Mozambique, only 37% of the population has access to basic sanitation as defined by WHO/UNICEF.”
2	In the study site description, the authors do not highlight access to water in this community yet it’s a key requirement for the pour-flush toilet technology.	This information was provided in Table 2 but we agree with the reviewer that it is important to flag it in the “setting” sub-section as well, so have made the point there.	Added clause (p. 5 line 10): “Though 99% have access to on-premises piped water”
3	The authors should clearly explain how they determined the sample size for the study. What is the 398? Is it compounds or individual participants. What was the study unit? And how did this impact their sample size calculation. All these aspects are not clear.	The reviewer is right that this was unclear. We have clarified in the manuscript that the sample size calculation is for the number of participants. We also clarified in Online Appendix B that if only one eligible respondent could be identified per compound, we moved onto the next compound. Therefore, there was no specific target sample size for compounds.	Clause amended (p. 8 line 6): “The sample size calculation for the number of participants to be surveyed...” Sentence added to Online Appendices p.7: “If only one eligible respondent could be identified on a compound, we moved onto the next compound.”
4	How did the authors ensure that participants in the control group were still using the same sanitary facility at the endline evaluation?	This was outlined in Online Appendix B. However, we have now also flagged it in the manuscript.	Amended sentence (p. 6 line 11) “...inspection of the toilet used to assess eligibility...”
5	What was the criteria for selecting control and intervention compounds?	The criteria for identifying intervention and control compounds in the MapSan trial were explained in the	n/a

	Reviewer 2	Response	Action taken
	The authors should clearly explain how the randomization was done.	“Study design” section. That section also explains that MapSan was a non-randomised trial with a controlled before-and-after design. Therefore, there was no randomisation. The study reported in this manuscript is observational, and the process for identifying compounds was set out in the “participants” sub-section.	

VERSION 2 – REVIEW

REVIEWER	Hennegan, Julie Johns Hopkins University Bloomberg School of Public Health, Environmental Health and Engineering
REVIEW RETURNED	08-Aug-2022

GENERAL COMMENTS	The authors' revisions have addressed my concerns and strengthened the paper. The paper is much clearer, and the limitations of the design have been highlighted appropriately for readers.
--